# Mucoadhesive Nanoparticles for Drug Delivery to the Anterior Eye

**DOI:** 10.3390/nano10071400

**Published:** 2020-07-18

**Authors:** Nicole Mangiacotte, Graeme Prosperi-Porta, Lina Liu, Megan Dodd, Heather Sheardown

**Affiliations:** Department of Chemical Engineering, McMaster University, 1280 Main St. W., Hamilton, ON L8S 4L7, Canada; nicole.mangiacotte@gmail.com (N.M.); graeme.prosperiporta@ucalgary.ca (G.P.-P.); liul5@mcmaster.ca (L.L.); Megan.J.Dodd@gmail.com (M.D.)

**Keywords:** Nanoparticles, HEMA, dexamethasone, eyedrops, drug release

## Abstract

While the use of topical drops for the delivery of drugs to the anterior of the eye is well accepted, it is far from efficient with as little as 5% of the drug instilled on the eye actually reaching the target tissue. The ability to prolong the residence time on the eye is desirable. Based on the acceptability of 2-hydroxyethyl methacrylate based polymers in contact lens applications, the current work focuses on the development of a poly(2-hydroxyethyl methacrylate (HEMA)) nanoparticle system. The particles were modified to allow for degradation and to permit mucoadhesion. Size and morphological analysis of the final polymer products showed that nano-sized, spherical particles were produced. FTIR spectra demonstrated that the nanoparticles comprised poly(HEMA) and that 3-(acrylamido)phenylboronic acid (3AAPBA), as a mucoadhesive, was successfully incorporated. Degradation of nanoparticles containing N,N′-bis(acryloyl)cystamine (BAC) after incubation with DL-dithiothreitol (DTT) was confirmed by a decrease in turbidity and through transmission electron microscopy (TEM). Nanoparticle mucoadhesion was shown through an in-vitro zeta potential analysis.

## 1. Introduction

HEMA is a water-soluble monomer that can easily be polymerized into a water insoluble polymer [1,2,3]. Typically poly(HEMA) hydrogels have a water content of approximately 40%, which can be varied to some extent by changing the crosslinking density [4,5]. Poly(HEMA) is biologically inert, resistant to degradation, has a high chemical stability, and is not damaged by high heat or pressure [6,7].

There are many potential applications for poly(HEMA) due to its similar density and water content when compared to living tissue [8]. HEMA was originally designed as an ophthalmic material and continues to be the most frequently used hydrophilic monomer in soft contact lenses [4]. Poly(HEMA) has also been used to prepare nanoparticles. The controlled release of several drugs, including hydrophilic anticancer drugs, from poly(HEMA) nanoparticles has been documented [9]. HEMA-based nanoparticles have also been modified with an adsorbent in order to purify human serum albumin, antibodies, and DNA [10,11,12]. In many cases, the HEMA particles are combined with other materials to incorporate a response to pH or temperature [13,14,15].

The major routes of drug delivery to the anterior eye are topical and subconjunctival [16]. Eye drops are the most common form of topical delivery and are attractive because they are self-administrable and non-invasive [17]. Eye drops may be made up of a variety of solutions and particle-based materials containing an ocular drug [18].

In all cases, however, this delivery modality must overcome the many evolutionary barriers that protect the eye from foreign substances including blinking, the tear film and its turnover rate, drainage, and induced lacrimation. The tear film has a turnover rate of 2 to 3 min and contains mucin that forms a hydrophilic layer for trapping debris and pathogens [19,20]. Due to the flow in the eye being directed to the nasal cavity and capillaries local to the conjunctival sac, 95% of the drug administered is removed systemically or via nasolacrimal drainage and only 5% or less is able to reach the intraocular tissues [17,19]. Additionally, the corneal epithelium acts as a mechanical barrier [17,19] limiting penetration to the internal ocular structures.

Improving drug delivery to the anterior eye can be achieved through modification of the drug or the delivery system. The type of formulation can be changed from an eye drop to a gel, ointment or insert [17]. Alternatively, the volume per dosage can be reduced to increase the local/systemic drug ratio [21]. Finally, the addition of mucoadhesive materials to the nanoparticles in the drops may help to extend the residence time in the eye [22]. There are several categories of mucoadhesive polymers including cationic, anionic, amphoteric, and boronic acid [17]. Boronic acid copolymers in particular have shown potential as mucoadhesive materials because of their interactions with the diols and sialic acid residues of mucin [23,24].

The overall objectives of this work revolve around the creation of novel ophthalmically compatible nano-sized polymer particles (<200 nm, ideally) to overcome the limitations of drug delivery to the eye. Thus, the particles should be able to load and release ophthalmic drugs and degrade under reducing conditions. To increase the residence time at the front of the eye, the nanoparticles should be mucoadhesive. This will allow an increased amount of drug to be released in the desired location.

## 2. Materials and Methods

### 2.1. Materials

HEMA, ethylene glycol dimethacrylate (EGDMA), inhibitor remover, methacrylic acid (MAA), sodium dodecyl sulfate (SDS), benzoyl peroxide (BPO), DTT, potassium bromide (KBr), 1-butanol, dexamethasone, 3AAPBA, bovine submaxillary mucin (BSM), and MTT reagent (3-(4,5-dimethylthiazol-2-yl)-2,5-diphenyl tetrazolium bromide) were purchased from Sigma-Aldrich. BAC was purchased from Alfa Aesar and potassium chloride (KCl) was purchased from EMD chemicals. Spectra/Por^®^ 6 regenerated cellulose 50 kDa molecular weight cut off (MWCO) dialysis tubing was purchased from Spectrum^®^ Laboratories. Acrodisc CR 13 mm high pressure liquid chromatography (HPLC) grade syringe filter with a 0.2 μm pore size were purchased from PALL Life Sciences.

### 2.2. Nanoparticle Preparation

#### 2.2.1. Poly(HEMA) Nanoparticle Preparation

Poly(HEMA) nanoparticle suspensions were prepared as follows. HEMA and EGDMA were passed through a column packed with inhibitor remover. The components of the organic phase, 9.30 mmol of HEMA, 0.795 mmol of EGDMA, 0.504 mmol of MAA, 0.186 mmol BPO, and 1.5 mL of 1-butanol, were mixed together in a 20 mL scintillation vial until the BPO dissolved. Next, a 200 mL 0.06% *w*/*v* SDS solution was prepared in a 500 mL Erlenmeyer flask and a magnetic stir bar was added. The flask was then sealed and bubbled with N_2_ gas for 40 min. The organic phase solution was added to the sealed flask. Next, the flask was placed in an oil bath on top of a heated stir plate and mixed at 700 rpm at a temperature of 80 °C for 2 h. The temperature was then increased to 90 °C for 1 h at which time the flask was removed from the heated oil bath and left to cool to room temperature.

#### 2.2.2. Modifications to Preparation Method

Formulations containing BAC, to incorporate degradability into the particles, were made using a method similar to the one described in the previous section with the exclusion of EGDMA from the organic phase, the addition of 0.595 mmol of BAC to the organic phase, and the increase in 1-butanol to 4 mL. Formulations containing 3AAPBA, for the incorporation of mucoadhesion, were made using a modified version of the method described in the previous section. The method was altered by the addition of 0.393 mmol of 3AAPBA to the organic phase. Drug loaded formulations were prepared as described above with the addition of 18 mg of dexamethasone to the organic phase.

#### 2.2.3. Purification of Nanoparticle Suspension

To purify the nanoparticle suspensions, the cooled products were placed in Spectra/Por^®^ 6 regenerated cellulose dialysis tubing with a MWCO of 50 kDa. The tubing was placed in 4 L plastic containers filled with milliQ water. The milliQ water was changed 10 times over the course of 10 days. After 10 days, the nanoparticle suspension was removed from the tubing, frozen, and freeze dried.

### 2.3. Nanoparticle Characterization

#### 2.3.1. Size Determination

The dynamic light scattering (DLS) function of the Brookhaven 90 Plus Particle Size Analyzer (50 Blue Point Rd, Holtsville, NY 11742, United States) was used to obtain the average effective diameters of the nanoparticles. A total of 1 mL of the nanoparticle sample was added to a polystyrene clear sided cuvette. Between 2 and 4 mL of MilliQ water was added to the cuvette in order to dilute the solution to obtain an acceptable level of transparency, which was determined by the suggested range for the count rate. The sample in the cuvette was well mixed and then analyzed by the Brookhaven Size Analyzer.

#### 2.3.2. Molecular Composition

The molecular structure of the nanoparticles was determined using Fourier Transform Infrared Spectroscopy (FTIR) with a Bruker Hyperion 3000 Microscope (40 Manning Road, Manning Park Billerica, MA 01821, USA) with a Vertex 70 Bench and HTS Plate Reader. Freeze dried poly(HEMA) nanoparticles crosslinked with EGDMA were analyzed using attenuated total reflectance FTIR (ATR-FTIR). Freeze dried poly(HEMA) nanoparticles co-polymerized with 3AAPBA and crosslinked with BAC were evenly dispersed in KBr powder. The mixture was packed into the well of a metal plate prior to analysis.

#### 2.3.3. Morphology

TEM was used to observe the morphology of the nanoparticles. First, the nanoparticle suspensions were diluted by a factor of 5 to 10 with milliQ water after which 5 μL of the diluted suspension was added to a TEM grid for TEM analysis. Analysis was performed using a JEOL 1200EX TEMSCAN (11 Dearborn Rd, Peabody, MA 01960, United States) at a magnification of 15,000 and 25,000.

#### 2.3.4. Degradation

To evaluate the degradation of the materials, the pH of MilliQ water was adjusted to 8.5 using 0.1 M sodium hydroxide. A 20 mM DTT solution was made in a 10 mL sealed round bottom flask, with the pH adjusted MilliQ water, and bubbled with N_2_ gas for 10 min. 4 mL of the nanoparticle suspension, poly(HEMA) crosslinked with BAC, was added to two separate 10 mL round bottom flasks. A total of 4 mL of pH 8.5 MilliQ water and 4 mL of the 20 mM DTT solution were added to the first and second flasks containing the nanoparticle suspension, respectively. Both nanoparticle flasks were bubbled with N_2_ gas for 10 min and placed in a shaking incubator at 37 °C for a minimum of five days (Figure 1).

An total of 300 μL of each solution was subsequently added to a Costar UV transparent 96-well plate. The absorbance values of the samples were obtained at 350 nm using a Tecan M200 Infinite Pro plate reader (9401 Globe Center Dr Suite 140, Morrisville, NC, USA). The solutions were also analyzed according to the morphology section.

#### 2.3.5. Mucoadhesion

The zeta potential function of the Brookhaven 90 Plus Particle Size Analyzer was used to assess the mucoadhesive properties of the nanoparticle formulations. Stock solutions of 100 mM KCl and 4 mg/mL BSM were prepared prior to zeta potential sample preparation. Test samples were prepared by adding 0.4 mL of a poly(HEMA, BAC, 3AABPA) nanoparticle sample or 0.1 mL of a poly(HEMA, BAC) sample, 0.2 mL of the BSM stock solution, and 0.5 mL of the KCl stock solution to a 2 mL Eppendorf, followed by diluting the sample to 2 mL with MilliQ water. Nanoparticle control samples were prepared in a similar manner to the test samples with the exception of the 0.2 mL of BSM stock solution being replaced with MilliQ water. Mucin control samples were also prepared in a similar manner to the test samples; however, the nanoparticle sample volume was replaced with MilliQ water. These samples were then placed in a shaking incubator at 37 °C for 3 h, at which time they were transferred to a cuvette and an AQ-1204 probe was inserted into the solution. The probe was then connected to the Brookhaven 90 Plus Particle Size Analyzer and the zeta potential of the samples was determined. A decrease in zeta potential should be observed when mucin adheres to the nanoparticle surface due to the negative zeta potential associated with mucin. In order to confirm mucoadhesion, the zeta potential of the nanoparticle samples incubated with mucin should be significantly more negative relative to the zeta potential of the mucin control and the relative nanoparticle control.

### 2.4. Drug Release Studies

To remove the loosely entrapped drug prior to drug release, 25 mL of the drug-loaded nanoparticle suspension was ultracentrifuged eight times with an increasing rotation speed between 10,000 rpm and 20,000 rpm. The pellet was removed and placed in a glass vial after each centrifugation step. The collection of pellets was then resuspended in 25 mL of milliQ water by sonication using a misonix S-4000 sonicator (1938 New Highway, Farmingdale, NY, USA) for 15 min. The resuspended drug-loaded nanoparticles were then put into a Spectra/Por^®^ 6 regenerated cellulose dialysis tubing (769 Jersey Ave, New Brunswick, NJ, USA) with a 50 kDa MWCO. The dialysis tube was then placed in a tube with 25 mL of MilliQ water, maintained at a temperature of 34 °C, and shaken continuously. The entire volume of water surrounding the dialysis tubing was removed and replaced at specified intervals to obtain drug release measurements and to ensure sink conditions. The collected samples were filtered using HPLC grade Acrodisc CR 13 mm syringe filters with a pore size of 0.2 μm. The filtered samples were analyzed using HPLC (232 Britannia Rd E, Mississauga, ON L4V 1S6, Canada) with a water/acetonitrile (60/40, *v/v*) mobile phase flowing at 1 mL/min, an Atlantis dC18 5 μm (6 × 100 mm) column, a Waters 1525 Binary HPLC pump, a Waters 2707 Autosampler, and a Waters UV/Visible Detector set to a wavelength of 254 nm. The first 60% of the drug release curves were analyzed using the Korsmeyer–Peppas model Equation (1).
(1)$$\frac{M_{t}}{M_{\infty }}=Kt^{n}$$
where *t* is the time selected, *M_t_* and *M*_∞_ are the mass released at time t and the amount of drug loaded, respectively, *K* is the rate constant, and *n* is the release exponent [20]. *K* and *n* were found using Equations (2) and (3)
(2)$$n=\frac{N\sum_{i=1}^{N}\left(ln\;t_{i}\times ln\;{\left(\frac{M_{t}}{M_{\infty }}\right)}_{i}\right)-\sum_{i=1}^{N}\left(ln\;t_{i}\right)\times \sum_{i=1}^{N}\left(ln\;{\left(\frac{M_{t}}{M_{\infty }}\right)}_{i}\right)}{N\sum_{i=1}^{N}{\left(ln\;t_{i}\right)}^{2}-{\left(\sum_{i=1}^{N}ln\;t_{i}\right)}^{2}}$$
(3)$$K=exp\left(\frac{\sum_{i=1}^{N}\left(ln\;{\left(\frac{M_{t}}{M_{\infty }}\right)}_{i}\right)-n\sum_{i=1}^{N}\left(ln\;t_{i}\right)}{N}\right)$$
where *i* refers to each drug release measurement and N is the total number of measurements used to calculate the coefficients [25]. These equations were obtained using the least squares fitting technique [25]. The release exponent and release constant are used to make assumptions regarding the drug release mechanism and the structural characteristics of the system, respectively [26]. After eight weeks, it was assumed that the entire amount of the drug loaded had been released from the nanoparticles. Therefore, drug loading was determined by measuring the amount of drug that was released after eight weeks.

### 2.5. Cell Viability Studies

The viability of human corneal epithelial cells (HCEC) in the presence of the nanoparticle suspensions was determined with an MTT assay. HCEC (10,000/well) and 200 μL of keratinocyte serum free medium (KSFM) were added to the wells of a 96-well microtiter plate (Grenier). The plate was stored in the incubator at 37 °C and 5% CO_2_ for a minimum of 3 h. Next, 50 μL of the nanoparticle suspensions, at original and reduced concentrations, was added to the wells containing HCEC. Then, the well plate was put into the incubator at 37 °C and 5% CO_2_ for 2 days. After 2 days, the media and nanoparticle suspensions were removed from the well plate, followed by the addition of 100 μL of KSFM and 10 μL of MTT reagent solution, at a concentration of 5 mg/mL in phosphate buffered saline (PBS), to each well. The cells were incubated with the MTT reagent at 37 °C and 5% CO_2_ for 2 h. During this incubation period, the water soluble MTT reagent was cleaved by intracellular succinate dehydrogenase resulting in formazan, which is unable to permeate through the membranes of healthy cells [27]. Then, the MTT reagent and media were removed and replaced with 50 μL of dimethyl sulfoxide (DMSO) to dissolve the formazan crystals. Finally, the amount of formazan was obtained by measuring the absorbance at a wavelength of 540 nm using a Tecan M200 Infinite Pro plate reader. The cell viability (%) relative to the controls, without nanoparticles, was determined by Equation (4)
(4)$$Cell\;Viability\;\left(\%\right)=\frac{{\left[A\right]}_{test}}{{\left[A\right]}_{control}}\times 100\%$$
where [*A*]_*test*_ and [*A*]*_control_* are the absorbance of the test well and the control well, respectively.

## 3. Results and Discussion

### 3.1. Nanoparticle Characterization

#### 3.1.1. Size Determination

Poly(HEMA) nanoparticles were analyzed using DLS to determine the diameter (Table 1). The diameters listed in the table were within the range of 97.8 nm to 125.5 nm. This range of diameters did not exceed the maximum of 200 nm specified in the objectives. Therefore, all formulations met the size objective.

The polydispersity values, determined from DLS, did not appear to follow any trend and were all within the range of 0.032 to 0.063.

Poly(HEMA, BAC, 3AAPBA) nanoparticles were analyzed using DLS to determine the average effective diameter. The results, shown in Table 2, were within the range of 179.5 nm to 219.4 nm. Samples C2, C3, C4, and C5 met the size objective, whereas samples C1 and C6 did not. However, the diameters of the latter samples only exceeded 200 nm by approximately 4 to 20 nm. In comparison, the diameter of the p(HEMA, BAC) particles was determined to be ~235 nm (data not shown).

Overall, all nanoparticles synthesized had an average effective diameter of less than 203 nm.

#### 3.1.2. Molecular Composition

The molecular composition of the nanoparticles was determined using ATR-FTIR. The FTIR spectrum found for poly(HEMA) nanoparticles is shown in Figure 2. Figure 2 confirms that HEMA was polymerized to produce poly(HEMA) based on the peaks at the wavenumbers listed in Table 3, which were obtained from the literature [28].

FTIR was also used to confirm the presence of 3AAPBA. As can be seen in Figure 3, the peak around 650 to 700 cm^−1^ corresponded to the out of plane aromatic C–H bending [29]. This peak was present in all samples with the exception of the poly(HEMA, BAC) NP sample. This was expected because the poly(HEMA, BAC) NP sample B2 had no aromatic rings. There should have been aromatic rings, however, if 3AAPBA was present.

#### 3.1.3. Morphology and Degradation

The morphology poly(HEMA) nanoparticles, prepared according to sample A2 in Table 1, can be observed in Figure 4. This TEM image shows that spherical nano-sized particles were successfully synthesized.

TEM images of poly(HEMA, BAC, 3AAPBA) nanoparticles Figure 5a,c also show that spherical nano-sized particles were synthesized. The nanoparticles were incubated with DTT to show their response to a reductive environment. DTT reduced BAC through a thiol-disulfide exchange reaction. TEM images of the nanoparticles incubated with DTT for 5 days were obtained in order to provide a comparison to their original shape and size. The TEM images of the nanoparticles after incubation with DTT can be found in Figure 5b,d.

It is evident based on Figure 5 that the poly(HEMA, BAC, 3AAPBA) nanoparticles decreased in size after incubation with DTT, presumably because the dithiol bonds in the crosslinker were cleaved. The fragments observed in Figure 5b also appear to be less uniform, implying that the nanoparticles are being degraded. The poly(HEMA, BAC, 3AAPBA) nanoparticles shown in Figure 5d are lighter relative to the same nanoparticles without DTT shown in Figure 5c. Additionally, darker randomly shaped spots can be seen near the edges of the lighter nanoparticles. These observations, although different than the ones obtained from the previous sample, support the theory that the nanoparticles degrade in the presence of reducing agents such as DTT.

Hydrolysis occurs through chain scission as a result of a water molecule being added to the polymer backbone [30,31]. The water in the body is able to interact with materials to different extents based on their affinity [31]. Hydrolysis is a relatively fast degradation process [32,33]. The anions and cations from salts in solution, which impact the environmental acidity and alkalinity, may further influence polymer degradation through hydrolysis and oxidation [34]. Oxidation is a relatively slow process that involves increasing a molecule’s oxidation state [31]. Reductive environments also exist within the body and are often created by oxidative stress [31]. Reductive degradation may occur within min to h in highly reductive environments [35].

The degradation of polymers is important because it promotes their removal in a safe and non-invasive manner [31]. Degradation products should ideally be easily eliminated from the body [36]. The chemistry of the polymer is the most significant factor that influences degradation because it dictates the stability of functional groups, chemical reactivity, affinity to water, and swelling behavior of the polymer [29]. The degradable linkage may be located in the polymer’s backbone, side chains, crosslinks, or any combination of these [31].

Further confirmation of nanoparticle degradation in the presence of DTT was obtained by measuring the turbidity of the samples. The turbidity of each of the control and test solutions was determined indirectly by measuring the absorbance of each sample at 350 nm. As shown in Figure 6, the absorbance, and therefore the turbidity, decreased for every poly(HEMA, BAC, 3AAPBA) nanoparticle formulation after incubation with DTT as expected. This further confirms the observations from the TEM images included in Figure 5.

#### 3.1.4. Mucoadhesion

Zeta potential values obtained from the mucin control, poly(HEMA, BAC) sample C4 from Table 2, and poly(HEMA, BAC, 3AAPBA) samples C2–C6 from Table 2 with and without mucin are shown in Figure 7. The mucin control showed that mucin has a negative zeta potential, as expected. The zeta potential of nanoparticles with mucin adsorbed to their surface is expected to decrease relative to the zeta potential of the nanoparticles without mucin.

The zeta potential of the poly(HEMA, BAC, 3AAPBA) sample C2 from Table 2 incubated with mucin was found to be significantly lower than its corresponding nanoparticle control but not the mucin control. It is assumed that test samples with zeta potential values that are more negative than the relative control samples, but less negative or the same as the mucin control, are assumed to not be mucoadhesive. This is because the decrease in zeta potential may be due to the addition of mucin to the sample and not interactions between the nanoparticles and mucin. There was no significant relationship found between the zeta potential of the poly(HEMA, BAC) sample incubated with mucin and its corresponding nanoparticle control or the mucin control. Test samples with zeta potential values that are not significantly different from the corresponding controls and the mucin control are also considered to not be mucoadhesive. The zeta potentials of the poly(HEMA, BAC, 3AAPBA) samples C3–C6 from Table 2 incubated with mucin were found to be significantly lower than their corresponding nanoparticle control and the mucin control. Test samples with zeta potential values more negative than the mucin control were assumed to be mucoadhesive. In this case, the decrease in zeta potential cannot be explained by the mixture of mucin with the nanoparticles instead of interactions between the two solutes. Therefore, it is assumed that a zeta potential significantly lower than the mucin control and the corresponding nanoparticle sample is due to interactions between the nanoparticles and mucin.

### 3.2. Drug Release Studies

The release of dexamethasone from the poly(HEMA) nanoparticle formulation A3 from Table 1 and poly(HEMA, BAC, 3AAPBA) nanoparticle formulation C3 from Table 2, was measured over the course of seven days. This time period was selected because it was highly probable that the nanoparticles would not remain at the front of the eye for more than seven days based on an estimated mucosal turnover rate of 12 to 24 h [32]. Figure 8 shows the dexamethasone release profile from poly(HEMA) nanoparticles. From this release curve, a rate constant of 0.001 and a release exponent of 0.607 were obtained. According to the literature, a release exponent value within the range of 0.43 to 0.85 from a spherical polymeric controlled delivery system corresponds to a drug release mechanism dictated by anomalous transport. This means that various types of phenomena, including diffusion and polymer swelling, are likely contributing to the release of the drug from the polymer spheres [26].

Figure 9 shows the dexamethasone release profile from poly(HEMA, BAC, 3AAPBA) nanoparticles. The modeling of drug release suggests that a two stage release of dexamethasone from these poly(HEMA, BAC, 3AAPBA) nanoparticles may be occurring, with a second burst following degradation (data not shown). However, on the eye, it would be anticipated that the particles would be cleared before this degradation induced release occurred.

Due to the fact that the Korsmeyer–Peppas model is based on the first 60% of the drug release, which is supposed to be the linear region, the calculations cannot use the actual value obtained for *M*_∞_. Instead, *M*_∞_ was selected as the amount of drug released after 144 h. The rate constant and release exponent obtained from the release curve in the previously mentioned figure were 0.002 and 0.586, respectively. This release exponent corresponds to a drug release mechanism dictated by anomalous transport [26].

The percentage of dexamethasone loaded in the nanoparticles was lower for the poly(HEMA, BAC, 3AAPBA) nanoparticles relative to the poly(HEMA) nanoparticles. A potential reason for this is that the mass percentage of the poly(HEMA) suspensions was greater or equal to three times the mass percentage of the poly(HEMA, BAC, PBA) suspensions. Additionally, the release exponents obtained from both release curves were very similar, with only a 3.5% difference relative to the release exponent from the data in Figure 8. This means that the drug release mechanism from both formulations of nanoparticles was likely to be similar. A potential reason for the similarity in drug release mechanism is that the main component of the nanoparticles, poly(HEMA), remained the same in both formulations. Therefore, the swelling characteristics of both formulations would be similar. Unlike the release exponents, the release constant obtained from the data in Figure 9 was twice the value of the release constant obtained from the data in Figure 8. Therefore, it is likely that there were differences in the structural characteristics of the two previously mentioned nanoparticle formulations. This structural difference can be attributed to the change in crosslinker, from EGDMA to BAC, and potentially the crosslink density. The crosslink density may have been reduced for the poly(HEMA, BAC, 3AAPBA) formulation due to a lower molar amount of BAC being incorporated relative to EGDMA.

### 3.3. Cell Viability Studies

The cytoxicity of the nanoparticle suspensions was tested using human corneal epithelial cells and an MTT assay. The nanoparticle formulations examined were poly(HEMA) sample A1 from Table 1 and poly(HEMA, BAC, 3AAPBA) samples C2 to C6 from Table 2. The poly(HEMA) nanoparticles were tested at four different concentrations: no dilution (4 mg/mL), 2× dilution (2 mg/mL), 4× dilution (1 mg/mL), and 8× dilution (0.5 mg/mL). The poly(HEMA, BAC, 3AAPBA) nanoparticles were only tested at two different concentrations: no dilution (1 mg/mL) and 2× dilution (0.5 mg/mL), because the original suspensions were approximately 4× more dilute than the poly(HEMA) nanoparticle suspensions. The results from the first MTT assay, shown in Figure 10, showed that the HCEC viability was 123.6% to 182.5% after incubation with poly(HEMA) nanoparticles. Based on the results obtained, there does not appear to be a significant relationship between HCEC viability and the concentration of the poly(HEMA) nanoparticles. Potential reasons for cell viability of over 100% are that the polymer in the formulation is increasing the enzymatic activity which is mistaken for increased viability or that the nanoparticles are interfering with the colour development of the reagent. It is not believed that the nanoparticle formulations tested were actually promoting HCEC growth, however, a live–dead assay or another equivalent assay would be required to confirm this.

The results from the second MTT assay, included in Figure 11, showed that the HCEC viability was 69.8% to 85.1% after incubation with poly(HEMA, BAC, 3AAPBA) nanoparticles. Based on the results obtained, there does not appear to be a significant relationship between HCEC viability and the concentration of the poly(HEMA, BAC, 3AAPBA) nanoparticles. Additionally, there does not appear to be a significant relationship between HCEC viability and the majority of the poly(HEMA, BAC, 3AAPBA) formulations tested. The exception to this is that poly(HEMA, BAC, 3AAPBA) sample 6 was associated with a significantly lower HCEC viability compared to poly(HEMA, BAC, 3AAPBA) samples 2 and 3. The lower HCEC viability may have been caused by the increased amount of 3AAPBA incorporated into the nanoparticles. Overall, these results show that the nanoparticles tested did not have a serious impact on the HCEC viability but that the presence of BAC on the surface of the particles may have negatively impacted cell growth and metabolism. Additional studies will be necessary to determine whether this phenomenon occurs in vivo as well.

## 4. Conclusions

Poly(HEMA) based nanoparticles containing BAC and 3AAPBA were synthesized. The incorporation of 3AAPBA in the nanoparticles was confirmed by FTIR. The degradation of nanoparticles crosslinked with BAC was observed by TEM and turbidity. TEM showed that the poly(HEMA, BAC, 3AAPBA) nanoparticles were reduced to smaller fragments and the turbidity, which was measured indirectly by absorbance of the nanoparticle suspensions, decreased. Both poly(HEMA) and poly(HEMA, BAC, 3AAPBA) nanoparticles were able to uptake and release dexamethasone over a one week period. The viability of HCEC after incubation with poly(HEMA) and poly(HEMA, BAC, 3AAPBA) nanoparticles was 123.6% to 182.5% and 69.8% to 85.1%, respectively. Poly(HEMA, BAC, 3AAPBA) samples C3 to C6 from Table 2 were found to be mucoadhesive due to the change in the zeta potential after incubation with mucin. Future experimentation should be focused on in vivo testing of the mucoadhesion of the poly(HEMA, BAC, 3AAPBA) nanoparticle formulations. These studies should be conducted to test the residence time of the poly(HEMA, BAC, 3AAPBA) nanoparticles at the front of the eye.

## Figures and Tables

**Figure 1 nanomaterials-10-01400-f001:**
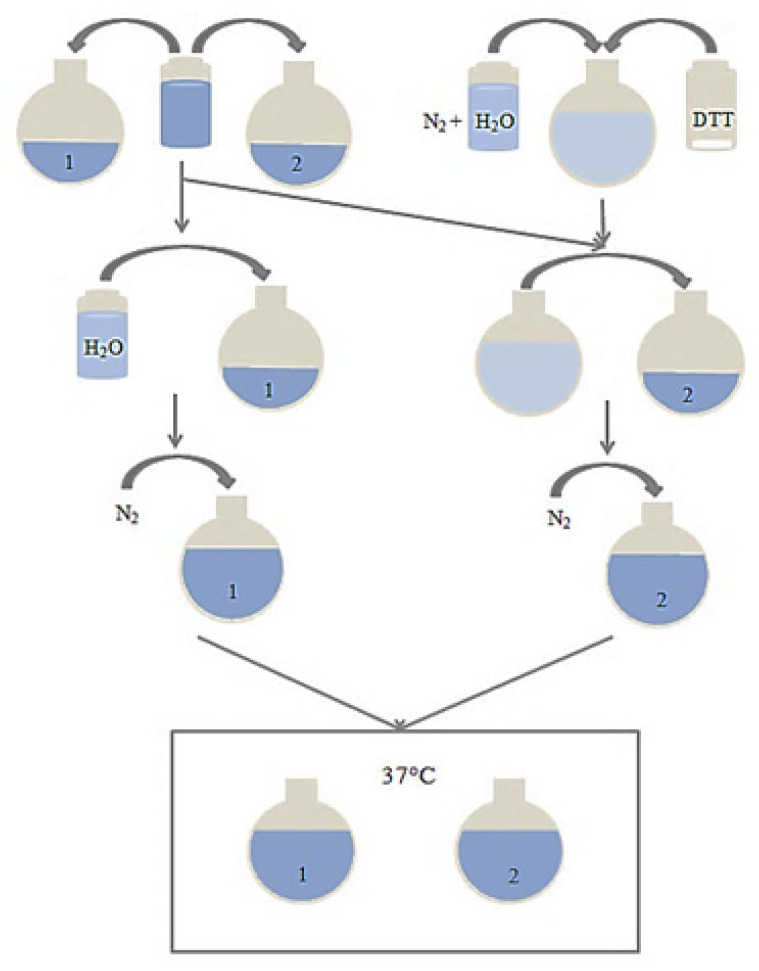
Schematic of method used for testing nanoparticle degradation.

**Figure 2 nanomaterials-10-01400-f002:**
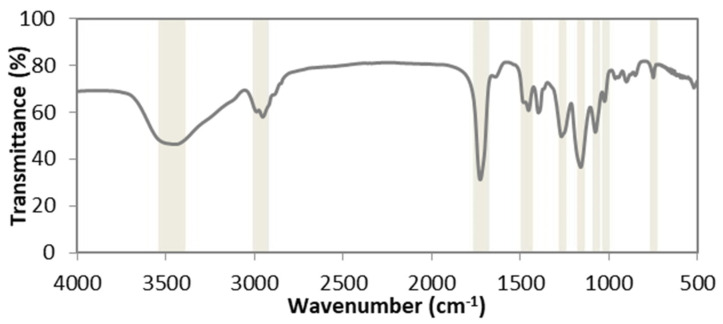
FTIR spectrum of poly(HEMA) nanoparticles crosslinked with EGDMA. The sample used to obtain the image was prepared according to the formulation listed for sample A1 in Table 1.

**Figure 3 nanomaterials-10-01400-f003:**
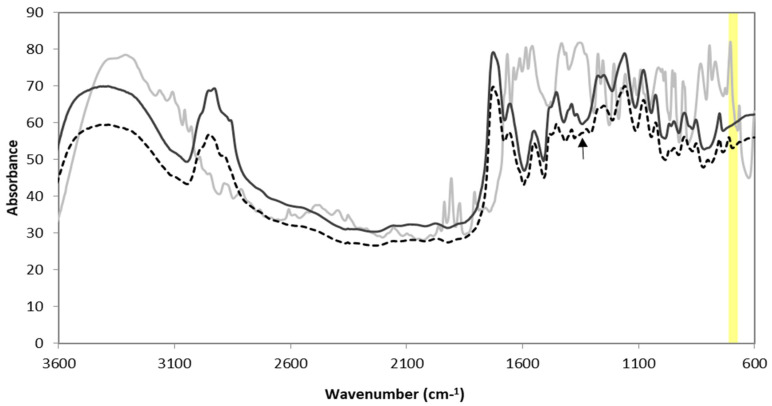
FTIR spectrum of 3AAPBA (▬), poly(HEMA, BAC) nanoparticles (▬), and poly(HEMA, BAC, 3AAPBA) nanoparticles (150) (**---**) (C6 fr—om Table 2).

**Figure 4 nanomaterials-10-01400-f004:**
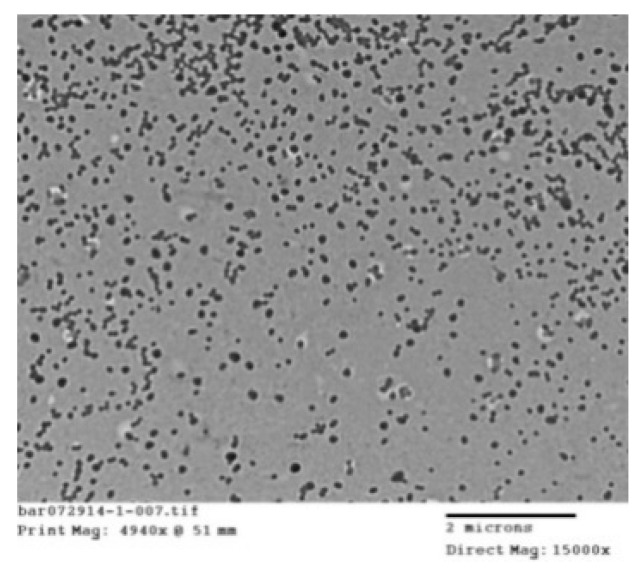
TEM image of poly(HEMA) nanoparticles prepared according to the formulation listed for sample A2 in Table 1.

**Figure 5 nanomaterials-10-01400-f005:**
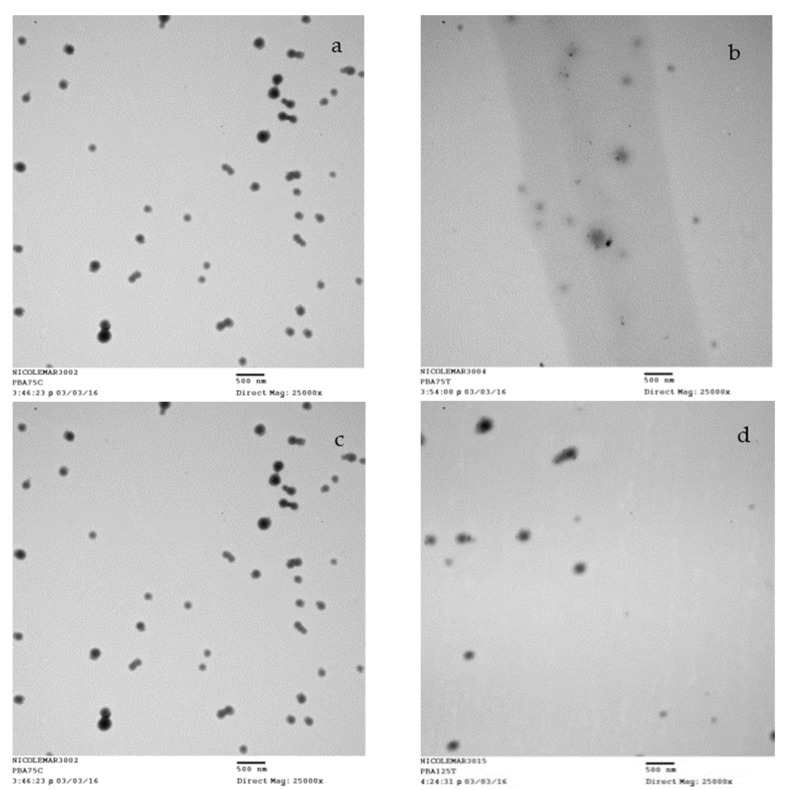
TEM images of samples after shaking incubation at 37 °C for 5 days. Samples shown are poly(HEMA, BAC, 3AAPBA) nanoparticles, sample C3 and C6 in Table 2, in the presence of water (**a**,**c**) and 10 mM DTT (**b**,**d**), respectively. For preparation, the samples were diluted and 5× and 5 μL was added to the TEM grid.

**Figure 6 nanomaterials-10-01400-f006:**
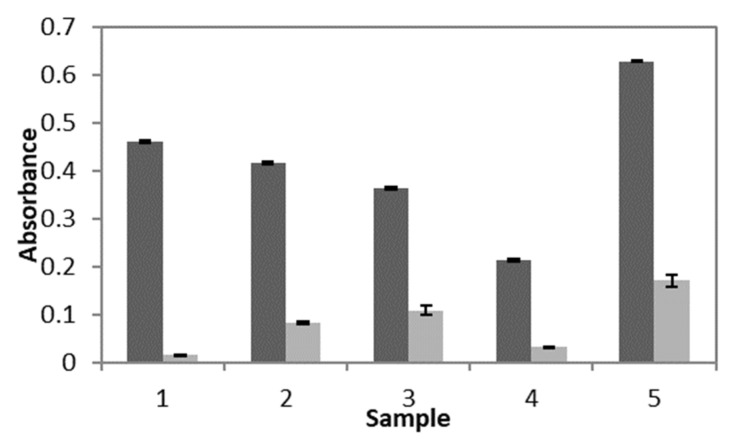
Absorbance readings at 350 nm of pHEMA (BAC, 3AAPBA) nanoparticles in the presence of water (■) and 10 mM DTT (■). Sample numbers correspond to samples C1 (1), C2 (2), C3 (3), C4 (4), and C5 (5) from Table 2. Error bars were obtained from the standard error of 9 control samples and 27 test samples. For all samples *p* < 0.0001.

**Figure 7 nanomaterials-10-01400-f007:**
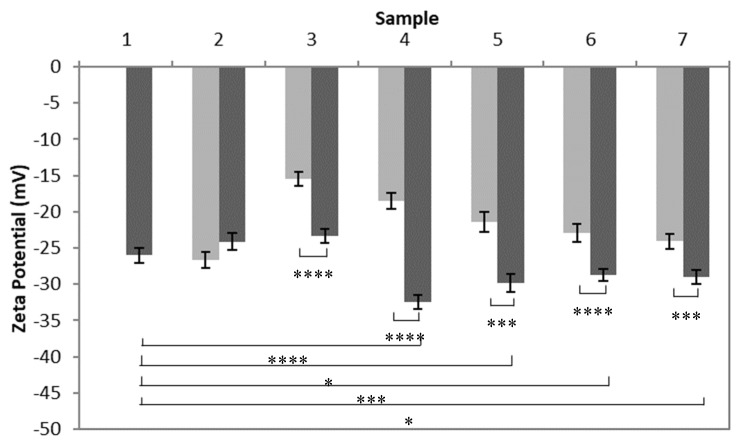
Zeta potential values of a mucin control solution (1), poly(HEMA, BAC) sample (2), and poly(HEMA, BAC, 3AAPBA) samples C2 (3), C3 (4), C4 (5), C5 (6), and C6 (7) from Table 3 with mucin (■) and without mucin (■). Error bars represent the standard error of 15 measurements. *p* value < 0.05 (*), <0.005 (***), and <0.0001 (****).

**Figure 8 nanomaterials-10-01400-f008:**
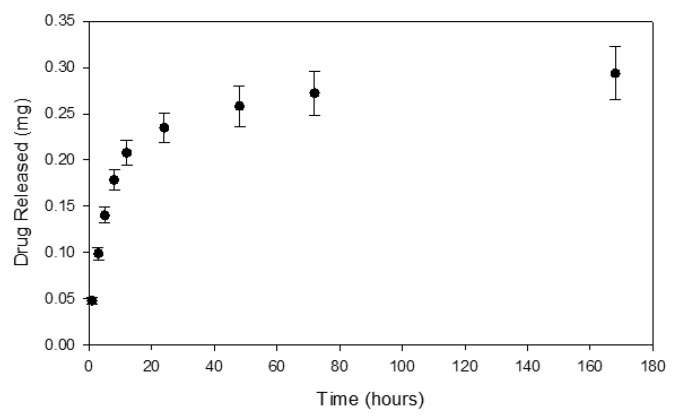
Dexamethasone release curve from loaded poly(HEMA) nanoparticles prepared according to sample A3 in Table 1 released from 50 kDa MWCO dialysis tubing under sink conditions. Initial loading of dexamethasone was 15.0% (±1.4%). Error bars were determined from the standard error.

**Figure 9 nanomaterials-10-01400-f009:**
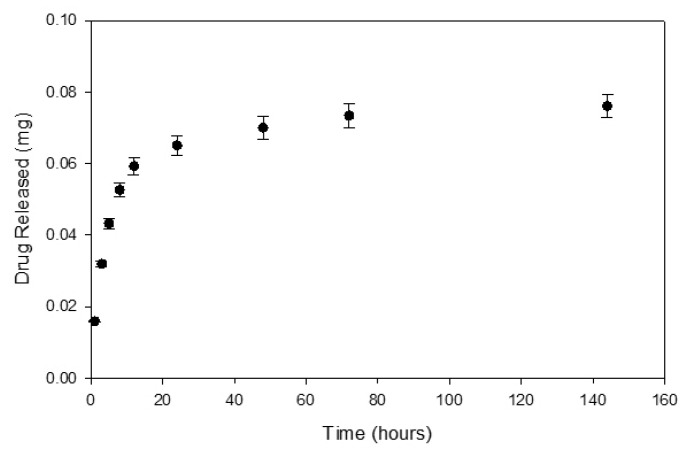
Dexamethasone release curve from loaded poly(HEMA, BAC, 3AAPBA) nanoparticles prepared according to sample C3 in Table 2 released from 50 kDa MWCO dialysis tubing under sink conditions. Initial loading of dexamethasone was 5.3% (±0.4%). Error bars were determined from the standard error.

**Figure 10 nanomaterials-10-01400-f010:**
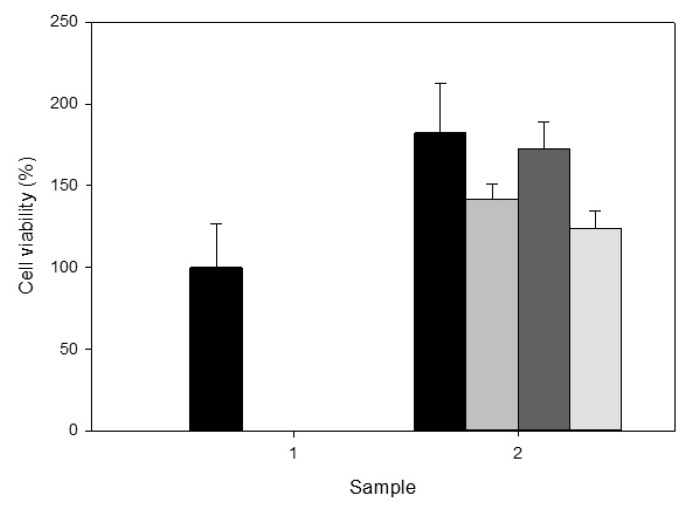
Cell viability of HCEC after incubation with poly(HEMA) and poly(HEMA, BAC) nanoparticles for 2 days at 37 °C and 5% CO_2_. Samples shown are the control with no nanoparticles (1) and poly(HEMA) nanoparticle formulation A1 (2) from Table 1. The bars correspond to original concentration (■), 2× dilution (■), 4× dilution (■), and 8× dilution (■). Error bars were obtained from the standard error of triplicate samples.

**Figure 11 nanomaterials-10-01400-f011:**
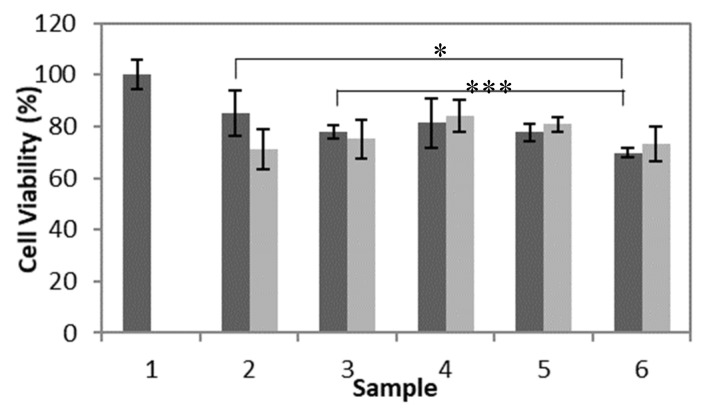
Cell viability of HCEC after incubation with poly(HEMA, BAC, 3AAPBA) nanoparticles Figure 2. days at 37 °C and 5% CO_2_. Samples shown are the control with no nanoparticles (1) and poly(HEMA, BAC, 3AAPBA) nanoparticle formulation C2 (2), C3 (3), C4 (4), C5 (5), and C6 (6) from Table 2. The bars correspond to original concentration (■) for all samples and 2× dilution (■). Error bars were obtained from the standard error of triplicate samples. *p* value < 0.05 (*) and <0.005 (***).

**Table 1 nanomaterials-10-01400-t001:** Average effective diameter and polydispersity of poly(HEMA) nanoparticles crosslinked with ethylene glycol dimethacrylate (EGDMA). Nanoparticle formulations contained various amounts of organic phase, surfactant, and monomer phase. The error associated with the diameter and polydispersity was obtained from the standard error of minimum triplicate measurements.

Sample	A1	A2	A3	A4	A5	A6	A7	A8
Monomer	0.75X				1X			
SDS (mg)	119		159		119		159	
1-Butanol (mL)	1.5	2.0	1.5	2.0	1.5	2.0	1.5	2.0
Average Diameter (nm)	107.5 ± 0.3	115.5 ± 0.5	97.8 ± 0.2	105.9 ± 0.3	125.5 ± 0.5	123.3 ± 0.5	119.6 ± 0.8	118.4 ± 0.4
Poly-Dispersity	0.032 ± 0.006	0.048 ± 0.005	0.031 ± 0.010	0.042 ± 0.017	0.063 ± 0.008	0.043 ± 0.012	0.054 ± 0.010	0.053 ± 0.016

**Table 2 nanomaterials-10-01400-t002:** Average effective diameter and polydispersity of poly(HEMA) nanoparticles crosslinked with BAC and copolymerized with 3AAPBA. Nanoparticle formulations contained various amounts of 3AAPBA. The monomer phase amount was kept constant at 0.75X and the BAC amount at 155 mg. The error associated with the diameter and polydispersity was obtained from the standard error of triplicate measurements.

Sample	C1	C2	C3	C4	C5	C6
3AAPBA (mg)	25	50	75	100	125	150
Average Diameter (nm)	219.4 ± 0.6	189.1 ± 0.9	183.8 ± 0.7	179.5 ± 0.7	182.8 ± 0.6	203.6 ± 0.9
Poly-Dispersity	0.093 ± 0.010	0.057 ± 0.010	0.054 ± 0.010	0.078 ± 0.010	0.086 ± 0.009	0.046 ± 0.009

**Table 3 nanomaterials-10-01400-t003:** Assignments for peaks at specified wavenumbers included in Figure 2.

Wavenumber (cm^−1^)	Assignment
3400–3500	OH stretching
2950	CH_2,_ CH_3_ stretching
1720	C=O
1450–1500	CH_2_ bending
1260	C–O stretching
1162	CH_3_ rocking, OH torsion
1074	O–C stretching (alcohol group)
1021	C–O stretching (ester group)
750	O=C–O stretching

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
