# Peer review of "Mucoadhesive Nanoparticles for Drug Delivery to the Anterior Eye"

_nanomaterials, 2020, doi:10.3390/nano10071400_

Round 1

Reviewer 1 Report

In this interesting paper, the authors describe the development and characterization of modified poly(HEMA) nanoparticles, designed to allow degradation and mucoadhesion, for a possible role to help achieving a more efficient drug delivery to the anterior of the eye.

The paper is well written and researched, the results are adequately described and discussed, I have only some minor remarks:

Line 176: the authors should explain what HCEC means, since the acronym is used here for the first time, and should list them in the Materials Section.

Line 207: I think the authors mean C6 instead of C5 toward the end of the line.

Line 214: I’m not sure where the 303 nm comes from, from the Tables it should e 203.6nm?

Line 233 and line 239: the sample depicted in Figure 4 is indicated as A2 in the text and A8 in the Figure caption, the authors should verify which is the case. I’d also suggest to increase the dimension of Figure 4.

Line 291: in the caption, the authors should add the indication of (2) after poly(HEMA, BAC) sample.

Line 345: could the author quantify the difference in the percentage of dexamethasone loaded in the two types of nanoparticles studied?

Line 360: this looks like an easy and efficient way to asses whether the authors’ hypothesis is true, maybe the authors could perform that equilibrium swelling test.

A final question: are there any reasons why different poly(HEMA) nanoparticles preparations have been used for the different experiments?

Author Response

The acronym HCEC has been defined. We thank the reviewer for pointing out this omission.

The reviewer is correct. The reference to C6 has been corrected.

The figure is now correctly identified. We thank the reviewer for pointing this out. While we agree that it would be useful to change the scale on the figure, unfortunately, due to the resolution on the original source photos we are not able to make this change. The size of the particles is included in the text. The figure is included mainly to demonstrate that the particles are spherical. This has been updated in the text. 

The caption in line 291 has been changed as requested.

The loading differences are noted in the text. We thank the reviewer for this helpful suggestion.

We agree that the hypothesis could be tested with a swelling test. However, this test, while quite straightforward to perform on bulk materials, is less accurate on particles and it was not possible to obtain this data. The reference to a swelling test has been removed from the text as a result.

The different preparations were used in different experiments to demonstrate the versatility of the process. A statement describing this rationale is provided in the revised manuscript.

Reviewer 2 Report

I find this is a very solid work which is of interest regarding the imoportant issue of topical drug delivery to the eye.

Regardless the authors will go on with this research, the work presented here has more than enough entity to be significant for an important number of readers of Nanomaterials.

However after reading the manuscript I have several comments that the authors can take into account:

1.- poly(HEMA, BAC) NPs characterization should be included, maybe as C0, in table 2 if available. The reason is that poly(HEMA, BAC) NP is an important material as it is the non-mucoadhesive control for comparison with the mucoadhesive particles. Further, the material appears several times in the text and Figures 3 and 7. In line 227, it appears even with a code B2 (B1 does not appear in the text).

2.- In figure 3 the Y axis should be absorption and not transmittance. Further I would suggest that the authors used the same kind of representation in figure 2 and 3, either transmittance or absorption.

3.- In figure 3 the region of interest corrresponds to 650-700 cm-1 as highlighted in yellow. It is difficult to see anything at that size, moreover with an overlay of 8 FTIR spectra. I think the authors should provide a zoom in that region next to the current Figure. There is enough space for it.

4.- In figure 5 the scale bar should be more visible with its correspondence in nm. There is a scale bar for each image that corresponds to 500 nm but it is hardly visible. In the caption, the authors provide the comment "The magnification of all images is 25000X": I would say this is not useful as that depends on the size of printing or displaying on a screen.

5.- I find the turbidity experiments (Figure 6) are much more reliable than TEM experiments to prove the degradability of the NPs. However, the authors should provide the time the NPs were incubated with DTT before the absorbace at 350 nm is measured for Figure 6. That parameter should be provided in the Figure caption or at least in the Experimental part, where it is just said for a minimum of 5 days. Is it not the same time for all the samples?

6.- I am curious why the degradation experiments were done with DTT, which is not present in biological media instead of reduced glutathion.

7.- In lines 325-326 the authors say: "It is important to note that the coefficients associated with the release curve in Figure 9 were calculated in a slightly different manner than stated in the methods." Which is that different manner?

8.- In lines 326-327 there is a sentence I don't understand: "The final amount of drug released was significantly higher than the amount released after one week."

9.- The mucoadhesive sample are usually named as poly(HEMA, BAC, 3AAPBA) and some others as poly(HEMA, BAC, PBA). The latter is less used, only from line 314 to 348.

Author Response

We agree that it was an omission not to include the characterization details for the BAC particles. The size of those particles has been added. 

We have modified the figure to include the relevant area to highlight the modifications. The y-axis title has been changed. We thank the reviewer for noticing this error.

We have removed the magnification comment from the figures. It is however not possible to alter the figures to change the scale bar. Higher resolution figures have been included and the size of the figures has been increased to better show the scale bar. The main purpose of these figures was to demonstrate that the particles prepared were spherical. The DLS data with sizes determined are shown to demonstrate the size differences between the different particles prepared.

We thank the reviewer for pointing out that the incubation time is not clear. All samples were incubated for 5 days as noted. This has been clarified in the manuscript.

DTT was chosen as initial experiments performed in glutathione showed slow degradation. Since the goal was to demonstrate that this degradation was possible, DTT was selected for these studies. This has been incorporated into the manuscript.

The comment about the modeling has been clarified in the manuscript. The release was as anticipated (diffusion based) early in the process for all samples. However the degradable samples showed a burst at the end as would be expected based on polymer degradation. This was heightened by the use of DTT. As a result, the total amount of drug released from these samples was higher than what was noted at the one week time point.

The nomenclature has been corrected. We thank the reviewer for making us aware of this.

Reviewer 3 Report

In this work, authors synthesized a poly(2-hydroxyethyl methacrylate(HEMA)) nanoparticles containing BAC and 3AAPBA for improving the residence time of the drug in the eye. The paper was well organized. Many experiments were implemented and analyzed. However, I think authors should address minor issues before it was published in this journal.

The author should explain whether the synthesized particles have health certification. If not, I think the authors should provide the biocompatible materials for proving the synthesized nanoparticles can fit human eyes.

Author Response

The reviewer has suggested that the materials be tested in vivo (by providing health certification). While we agree that this is optimal, this work is outside of the scope of the current project. As noted in the manuscript, all of the materials used in this work have been previously used on eye. Therefore, we do not envision significant toxicity issues. In a follow up publication, we will examine the loading and release of relevant drugs from these materials and will provide detailed in vivo results.